# N Absorption, Transport, and Recycling in Nodulated Soybean Plants by Split-Root Experiment Using $^{15}$N-Labeled Nitrate

Maria Doi [1], Kyoko Higuchi [1], Akihiro Saito [1], Takashi Sato [2] and Takuji Ohyama [1,*]

1    Department of Agricultural Chemistry, Tokyo University of Agriculture, Tokyo 156-8502, Japan
2    Faculty of Bioresource Sciences, Akita Prefectural University, Akita 010-0195, Japan
*    Correspondence: to206474@nodai.ac.jp

**Abstract:** Nitrate concentration is variable in soils, so the absorbed N from roots in a high-nitrate site is recycled from shoots to the root parts in N-poor niche. In this report, the absorption, transport, and recycling of N derived from $^{15}$N-labeled nitrate were investigated with split-root systems of nodulated soybean. The $NO_3^-$ accumulated in the root in 5 mM $NO_3^-$ solution; however, it was not detected in the roots and nodules in an N-free pot, indicating that $NO_3^-$ itself is not recycled from leaves to underground parts. The total amount of $^{15}NO_3^-$ absorption from 2 to 4 days of the plant with the N-free opposite half-root accelerated by 40% compared with both half-roots that received $NO_3^-$. This result might be due to the compensation for the N demand under one half-root could absorb $NO_3^-$. About 2–3% of the absorbed $^{15}$N was recycled to the opposite half-root, irrespective of N-free or $NO_3^-$ solution, suggesting that N recycling from leaves to the roots was not affected by the presence or absence of $NO_3^-$. Concentrations of asparagine increased in the half-roots supplied with $NO_3^-$ but not in N-free half-roots, suggesting that asparagine may not be a systemic signal for N status.

**Keywords:** soybean; split-root; $^{15}$N; transport; amino acids; ureides; gene expression

## 1. Introduction

Nitrogen (N) is one of the essential macronutrients required for plants, and the availability of N is often a limiting factor for plant growth and crop yield. Plant roots absorb N primarily in the forms of nitrate ($NO_3^-$) and ammonium ($NH_4^+$). Leguminous plants such as soybean incorporate atmospheric dinitrogen ($N_2$) in root nodules, forming a symbiosis with $N_2$-fixing rhizobia. Nitrate absorbed by soybean roots can be directly translocated to the shoot or first assimilated then transported mainly as asparagine (Asn) via xylem vessels to the leaves. Some parts of N assimilated in the leaves may be re-translocated to the apical part of the roots or buds via the phloem to support the N requirement of the developing organs [1]. Therefore, recycling N from leaves to the roots is important for young roots and nodules, but the mechanisms of how to control the recycling of N and how the N conditions around the roots affect N recycling are not fully understood.

As nitrate concentrations in the field vary by several order of magnitude, both spatially and temporally [2,3], plant roots need to regulate $NO_3^-$ absorption mediated by nitrate transporters [1]. The nitrate reductase (NR) activity is also finely regulated by various internal compounds and environmental conditions at the transcription, translation, protein modification, and degradation levels [4]. Nitrates and metabolites such as glutamine (Gln), sucrose, cytokinin, and light are regulatory factors for NR [4]. In addition, nitrate absorption is regulated systemically through root–shoot communications [2].

The fixed N in the nodule also recycles from the shoot to the roots. Oghoghorie and Pate [5] investigated the N transport of nodulated field pea (*Pisum arvense* L.) using $^{15}$N-labeled $N_2$ and $^{15}NO_3^-$. When the nodules at the upper roots were exposed to $^{15}N_2$, an appreciable amount of $^{15}$N translocated to the lower part of the roots. However, $^{15}$N

recycling was much lower when the shoot being removed just before the $^{15}N_2$ exposure. This result indicates that a part of fixed N in the nodules is recycled via shoots to the lower part of the roots, and little directly transported from nodules to the lower roots. The same was true when $^{15}NO_3^-$ was applied from the upper part of the lateral roots, some $^{15}N$ recycled from the shoot to the distal root part. Tanaka et al. [6] investigated the effect of various concentrations of $NO_3^-$ supply to one half-root of soybean by the split-root systems on the nodule growth, $N_2$ fixation, and the $^{15}N$ recycling of another half-root with N-free media after 20 or 40 days of treatment. When the half-roots were directly in contact with over 100 mg N $L^{-1}$ of nitrate, the nodule dry weight and acetylene reduction activity of the nodules were severely depressed, but the nodules attached to the opposite half-roots in N-free conditions were promoted. The effect on nodulation by the localized $NO_3^-$ supply to the split-root systems was investigated in the Williams parent and its hypernodulation mutant NOD1-3. The nodulation was inhibited only at the half-roots with $NO_3^-$ application but not in the N-free half-roots, irrespective of nodulation types [7]. Similar local repression at $NO_3^-$ site has been reported in *Trifolium repens* [8], *Casuarina* [9], and *Medicago truncatula* [10] by split-root systems, although the nitrogenase activity of root nodules on both half-roots with +N and −N was inhibited in the case of *Arachis hypogaea* [11].

When one-half of the sprit roots were in N-deficiency, the $NO_3^-$ absorption in the other half-roots supplied with $NO_3^-$ was promoted due to compensation of limited N absorption site. In this research, we investigated the absorption, transport, and recycling of $^{15}N$ originating from one half-root supplied with $^{15}NO_3^-$ to the shoots and another half-root supplied with N-free solution or non-labeled $NO_3^-$ using split-root systems at 2 and 4 days of treatments. The concentrations of major N metabolites, such as Asn, Gln, glutamate (Glu), aspartate (Asp), allantoin, and allantoate in each part of soybean plants were determined.

The main questions are as follows: Do the half-roots with N-free conditions accelerate the nitrate uptake in the other half-roots with $^{15}NO_3^-$, compared with both half-roots with $NO_3^-$? Is $NO_3^-$ itself transferred to the opposite side of the half-roots with N-free solution from the half-roots supplied with $NO_3^-$? Are the percentage of recycled-N in the half-roots affected by the N conditions? Do the N compounds such as Asn, Gln, or ureides accumulate in the half-roots with N-free solution affected by the other half-roots supplied with or without $NO_3^-$?

## 2. Materials and Methods

### 2.1. Split-Root Experiment Supplied with $^{15}N$-Labeled Nitrate from One Half-Root

Soybean seeds (*Glycine max* (L.) Merr., cultivar 'Williams') were surface-sterilized and sown in a vermiculite bed. At 7 days after planting (DAP), plants in a vermiculite bed were inoculated with a suspension of *Bradyrhizobium diazoefficiens* (USDA 110). At 10 DAP, the primary root below 5 cm long was cut off and the plant transplanted into an 800 mL of nitrogen-free nutrient solution [12] in a 900 mL glass bottle with continuous aeration. The bottle covered with aluminum foil for shading a culture solution. Plants were cultivated in a biophotochamber (LH-350S, Nippon Medical & Chemical instruments Co., Ltd., Osaka, Japan) under 28 °C day/18 °C night temperatures, 55% relative humidity, and under a photoperiod of 16 h light (228 μmol photons $m^{-2}$ $s^{-1}$)/8 h darkness.

At 24 DAP, each plant transferred to a two-compartment container (Figure 1). Two polyethylene terephthalate pots with a capacity of 1 L were attached. Each pot was filled with 0.8 L of N-free culture solution, covered with aluminum foil, and aerated continuously. Three treatments were imposed on 31 DAP using $NaNO_3$ (30.1 atom%$^{15}N$). The N0-N0 treatment: [L]; N-free, [R]; N-free. $^{15}N5$-N0 treatment: [L]; $^{15}N$-labeled 5 mM $NO_3^-$, [R]; N-free; $^{15}N$-N5 treatment: [L]; $^{15}N$-labeled 5 mM $NO_3^-$, [R]; non-labeled 5 mM $NO_3^-$. The 800 mL of 5 mM $NaNO_3$ solution contained 56 mg N per pot. At 2 day and 4 day treatments, plants were harvested, washed, and separated into the lower half of the noduled roots, which were in direct contact with the solution, the upper half of the noduled roots,

which were out of the solution, the basal part of roots, and the shoot. No nitrate ion was detected in the N-free solution after 4 days of cultivation in split-root culture, analyzed by the ion chromatograph (PIA-1000, Shimadzu Corporation, Kyoto, Japan), so mixing of the solution between two adjacent pots was excluded. The plants were dried using a freeze-dryer (VD-400F, TAITEC, Saitama, Japan) and separated into the leaves, stems plus petioles, basal roots, the upper part of nodules and roots in [L] or [R], and the lower part of nodules and roots in [L] or [R]. Then, the dry weight of them was measured and ground into a fine powder. About 2–5 mg of ground sample was packed in a tin capsule, and the N concentration and $^{15}$N concentration were analyzed by the elemental analyzer (Flash EA1112; Thermo Electron, Milan, Italy) coupled to an isotope-ratio mass spectrometer (Delta Plus XP; Thermo Fisher Scientific, Bremen, Germany). The percentage of N derived from $^{15}$N-labeled $NO_3^-$ ($\%^{15}$N) was calculated from the equation: $100 \times$ atom%excess of sample/atom%excess of labeled $^{15}NO_3^-$ (29.7 atom%excess in this experiment). The amount of N from $^{15}$N-labeled $NO_3^-$ was calculated by N content $\times \%^{15}$N/100. The distribution of N from $^{15}NO_3^-$ in each part was calculated based on the total N content from $^{15}NO_3^-$. The experiment was carried out with four biological replications.

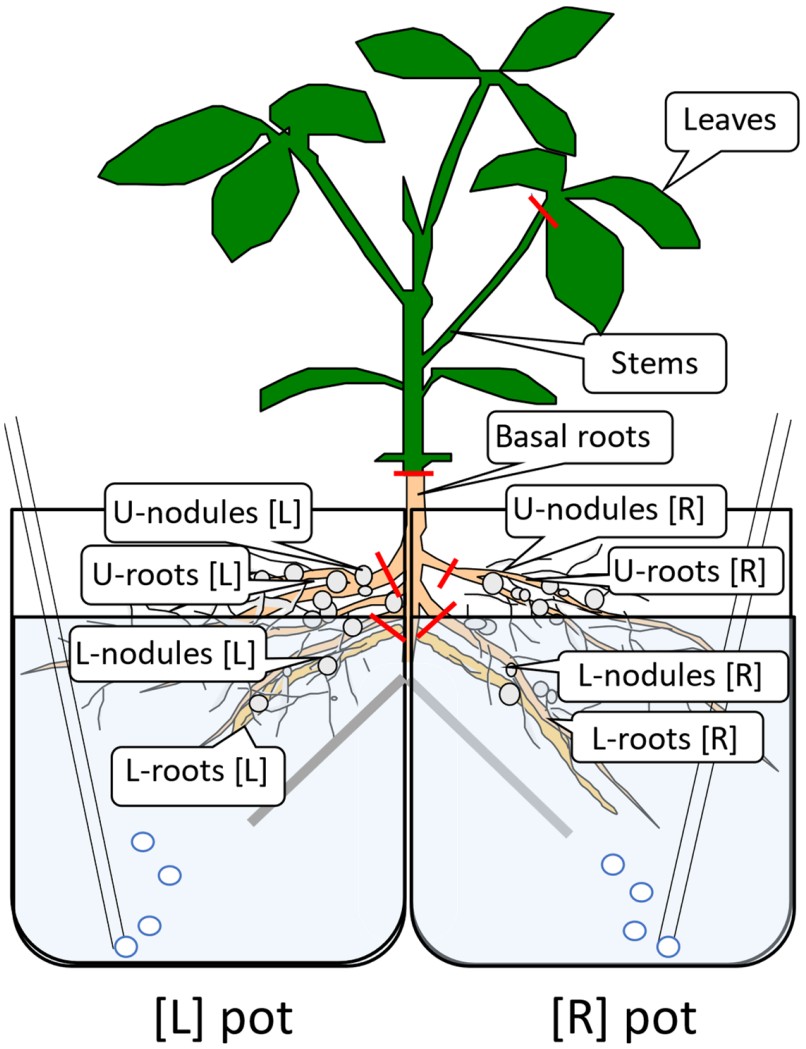

**Figure 1.** Split-root experimental setting. Three treatments were imposed, N0-N0, $^{15}$N-N0, and $^{15}$N-N5. In N0-N0 treatment, both pots contained N-free solution. In $^{15}$N-N0, and $^{15}$N-N5 treatments, the culture solution with $^{15}$N-labeled 5 mM $NO_3^-$ was supplied from the pot on the left [L], and either N-free ($^{15}$N-N0) or non-labeled 5 mM $NO_3^-$ ($^{15}$N-N5) was supplied in the pot in the right [R]. Both pots were continuously aerated. Plants were harvested at 2 days (2 DAT) and 4 days (4 DAT) of treatments. Plant parts separated at the red line.

## 2.2. The Effects of Split-Root Treatments on the Principal N Metabolites in Each Part of Soybean Plants

Freeze-dried plant powder was extracted with 80% ethanol containing 0.2 mM MES as an internal standard for capillary electrophoresis. The extracts were dried in vacuo and redissolved in water. Then, the concentrations of nitrate, Glu, Asp, Gln, Asn, allantoin, and allantoate were analyzed by capillary electrophoresis (7100, Agilent Technologies, Inc., Santa Clara, CA, USA) using a fused silica tube (inner diameter (id); 50 μm, 104 cm long) and a commercial buffer solution (α-AFQ109, Ohtsuka Electronics Co., Ltd., Osaka, Japan), with an applied voltage of −25 kV. Peaks detected with a signal wavelength of 400 nm and a reference wavelength of 265 nm. As half-roots in the right pot [R] and the left pot [L] were equivalent in N0-N0 and $^{15}$N5-N5 treatments, data of the concentrations for these treatments were combined (N = 8).

## 3. Results

### 3.1. $^{15}$N Translocation from the Half-Roots Supplied with $^{15}NO_3^-$ to the Other Parts

3.1.1. N Content in Each Part of Soybean Plants

Total amount of N in plant and the N content in each part at 2 DAT and 4 DAT were compared among N0-N0, $^{15}$N5-N0, and $^{15}$N5-N5 treatments to see the effect of $NO_3^-$ applications in one or both sides of half-roots (Figure 2). Total amount of N in the plant with N0-N0 was 30.1 mg N and 50.5 mg N at 2 DAT and 4 DAT, respectively. By the N0-N0 treatment, plants depended on sole nitrogen fixation, and the 20.4 mg increase of total N from 2 DAT to 4 DAT was derived from $N_2$ fixed by the root nodules. The increase in N content of the leaves from 2 DAT to 4 DAT was about 15 mg N, and the N contents were statistically significant ($p < 0.01$) between 2 DAT and 4 DAT based on Student's T-test. The N contents in the stems and basal roots tended to increase from 2 DAT to 4 DAT but were not significant. The N contents in both the upper and lower roots and nodules did not change significantly. The total amount of N in the plants with $^{15}$N5-N0 treatment was 29.5 mg N and 48.8 mg N at 2 DAT and 4 DAT, respectively. The N increase from 2 to 4 days was 19.3 mg N, almost the same as that in the N0-N0 treatment. The increase in the N content of leaves was 12 mg N, most remarkable among organs, and the N contents in the basal roots and the lower roots in contact with $^{15}NO_3^-$ were also significantly increased from 2 DAT to 4 DAT. Total amount of N in the plants with $^{15}$N5-N5 treatment was 32.5 mg N, and 53.5 mg N, at 2DAT and 4 DAT, respectively. Total amounts in $^{15}$N5-N5 treatment were slightly higher than those of N0-N0, and $^{15}$N5-N0 treatments, although statistically not significant. In this treatment, the increase in the total N content from 2 DAT to 4 DAT was 21 mg N, almost the same as those in N0-N0 (20.4 mg N) and $^{15}$N5-N0 (19.3 mg N) treatments. The N contents in the leaves, basal roots, and lower roots were significant between 2 DAT and 4 DAT. In all treatments, the increase of N in leaves was remarkable from 2 DAT to 4 DAT because the leaves grew faster in this stage (about V4), and the N fixation activity in this stage might become higher.

3.1.2. N Content from $^{15}$N-Labeled Nitrate in Each Part of Soybean Plants

To see how much $^{15}NO_3^-$ was absorbed and how the N derived from $^{15}NO_3^-$ was distributed among organs from 2 to 4 DAT, the N content from $^{15}NO_3^-$ in each part of plants with $^{15}$N5-N0 and $^{15}$N5-N5 treatments is shown in Figure 3. The total amounts of N derived from $^{15}NO_3^-$ were 2.51 mg N at 2 DAT and 10.4 mg N at 4 DAT in $^{15}$N5-N0 treatment, while those were 3.04 mg N at 2 DAT and 8.67 mg N at 4 DAT in $^{15}$N5-N5 treatment. The amounts of N derived from $^{15}NO_3^-$ in $^{15}$N5-N0 treatment at 4 DAT were significantly ($p < 0.05$) higher than that of $^{15}$N5-N5 treatment, suggesting that when N was applied only from one side of the split-roots, the N absorption might be accelerated compared with when both sides of the roots were supplied with $NO_3^-$. In both cases, $^{15}NO_3^-$ absorption was higher during the second 2 days period from 2 DAT to 4 DAT compared with the first 2 days from 0 DAT to 2 DAT. Total amount of N from $^{15}NO_3^-$ from 2 DAT to 4 DAT was 7.89 mg N and 5.63 mg N in $^{15}$N5-N0 and $^{15}$N5-N5 treatments, respectively. These

accounted for a 40% increase in one-side $NO_3^-$ supply ($^{15}$N5-N0) compared with both sides supplied with $NO_3^-$ ($^{15}$N5-N5). About half of $^{15}$N was distributed in leaves at 4 DAT with both $^{15}$N5-N0 (6.16 mg N) and $^{15}$N5-N5 (4.79 mg N) treatments. The accumulation of labeled-N in the leaves of the plants with $^{15}$N5-N0 treatment was significantly higher than that with $^{15}$N5-N5 treatment at 4 DAT ($p < 0.05$). The distribution of $^{15}$N-labeled N was also high in the lower roots, which were in direct contact with $^{15}$NO$_3^-$. Those were 2.36 mg N with $^{15}$N5-N0 treatment and 1.96 mg N with $^{15}$N5-N5 treatment. At the beginning of treatments, the $^{15}$N-labeled solution contained 56 mg N of nitrate per pot, and the total N absorption from $^{15}$NO$_3^-$ did not exceed 12 mg N at 4 DAT, so nitrate in the solution has not been depleted during the 4 days of $^{15}$N treatment.

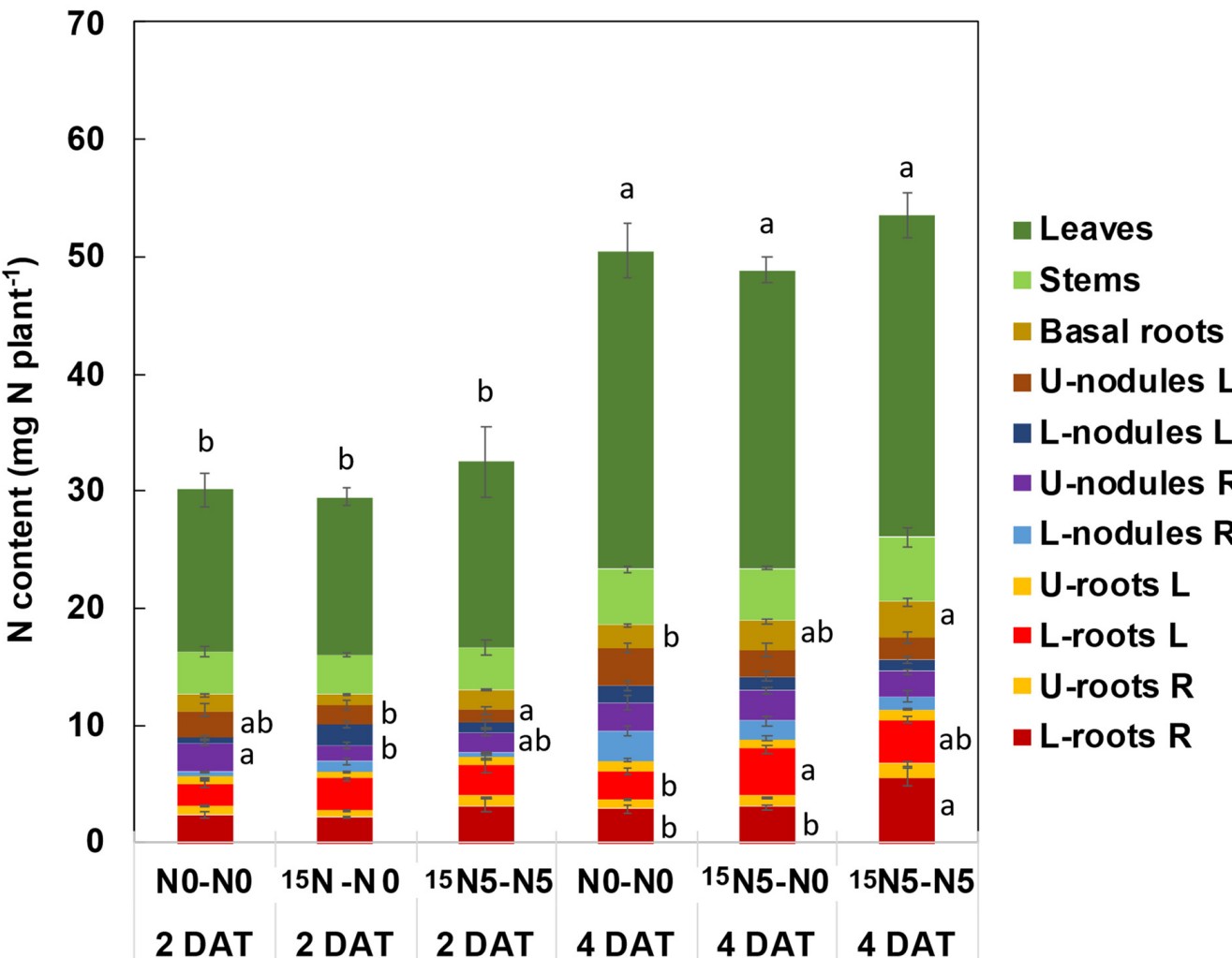

**Figure 2.** Nitrogen content in each part of soybean plants with after 2 and 4 days of N0-N0, $^{15}$N5-N0, and $^{15}$N5-N5 treatments. Different letters on the top of the columns indicates the significantly different in total N among treatments based on Tukey's Test ($p < 0.05$). Different alphabet at the side of the column indicates the significant difference of N content in the parts among treatments. The letter omitted in the part with no significant difference among three treatments. N = (4). Average ± standard error.

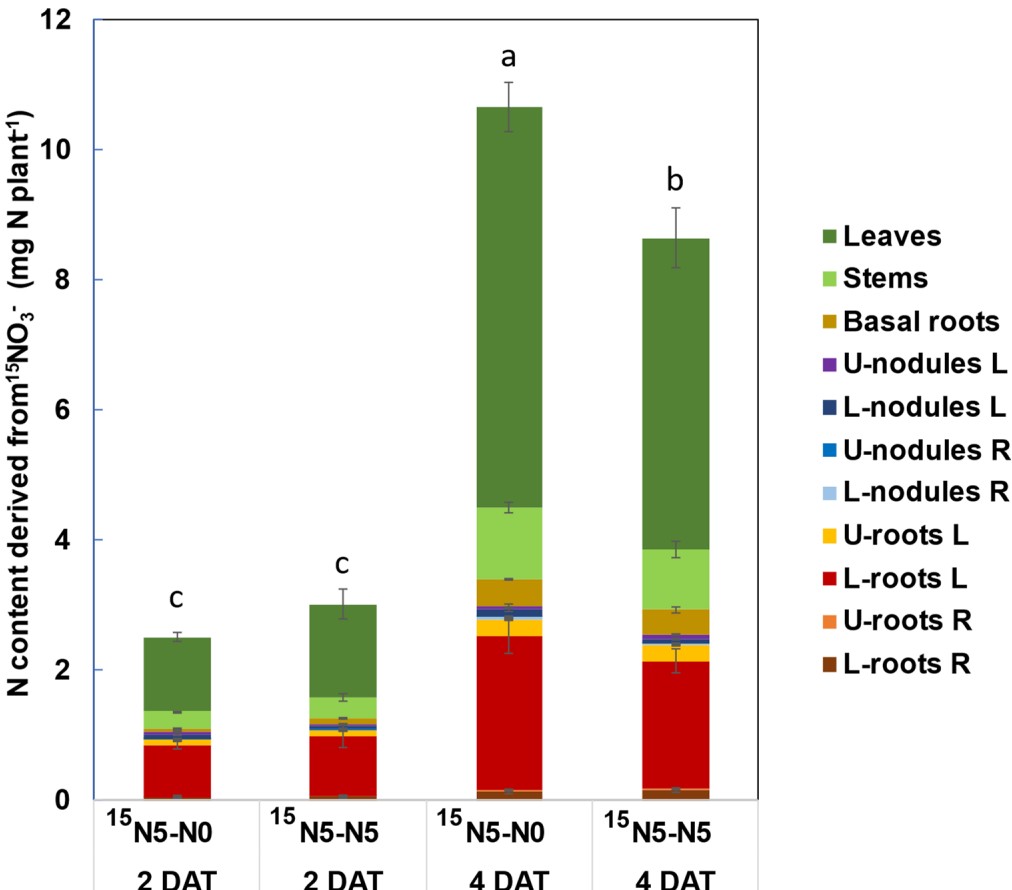

**Figure 3.** Nitrogen content in each part of soybean plants treated with $^{15}NO_3^-$ after 2 and 4 days of treatment. The experimental setting is described in the legend of Figure 2. Different letters indicate a significant difference ($p < 0.5$) based on Tukey's test. N = (4). Average $\pm$ SE.

The N derived from atmospheric N (Ndfa) and N derived from nitrate (Ndfn) were estimated by subtracting the amount of N derived from $^{15}NO_3^-$ from the increase in total N from 2 DAT to 4 DAT. The Ndfa from 2 DAT to 4 DAT in $^{15}$N5-N0 treatment was 13.14 (19.30 − 6.16) mg N. While the amount of Ndfn from $^{15}$N from 2 DAT to 4 DAT in $^{15}$N5-N5 treatment was 5.63 (8.67 − 3.04) mg N from the half-root with $^{15}NO_3^-$, both sides of the half-roots were in the same N conditions except for labeling, so the same amount of N was absorbed from non-labeled half-roots. Therefore, a total of 11.26 mg (5.63 + 5.63) mg N should derive from both the $^{15}$N-labeled and non-labeled nitrate. The total N increase from 2 DAT to 4 DAT was 21.0 mg N, so the amount of Ndfa was 9.7 (21.0 − 11.3) mg N in the $^{15}$N5-N5 treatment. The percentage dependence on Ndfa and Ndfn was 100% Ndfa and 0% Ndfn in N0-N0 treatment, 59% Ndfa and 41% Ndfn in $^{15}$N5-N0 treatment, and 46% Ndfa and 54% Ndfn in $^{15}$N5-N5 treatment.

### 3.1.3. Percentage of N Derived from $^{15}$N-Labeled Nitrate in Each Part of Soybean Plants

The percentage of N derived from $^{15}NO_3^-$ ($^{15}$N%) in total N in each part of the soybean plant with split-root systems is shown in Figure 4. The $^{15}$N% of leaves, stems, and basal roots were about 8.6, 8.0, and 5.8, respectively at 2 DAT, and there were no significant differences between $^{15}$N5-N0 and $^{15}$N5-N5 treatments. However, the $^{15}$N% of these organs at 4 DAT were significantly higher in the $^{15}$N5-N0 treatment, compared with the $^{15}$N5-N5 treatment (Figure 4A–C). It is the same for the $^{15}$N% of the lower roots, upper roots, lower nodules, and upper nodules in the non-labeled pot [R] between $^{15}$N5-N0 and $^{15}$N5-N5 treatments (Figure 4D–G). On the other hand, the $^{15}$N% of the roots and nodules in the $^{15}$N-labeled pot [L] were not significantly different between $^{15}$N5-N0 and $^{15}$N5-N5

treatments at 4 DAT except for the upper roots (Figure 4H–K). These results might suggest that the half-roots supplied with N-free solution accumulated more $^{15}N$ derived from the other half-roots with 5 mM $^{15}NO_3^-$ than those with 5 mM non-labeled $NO_3^-$.

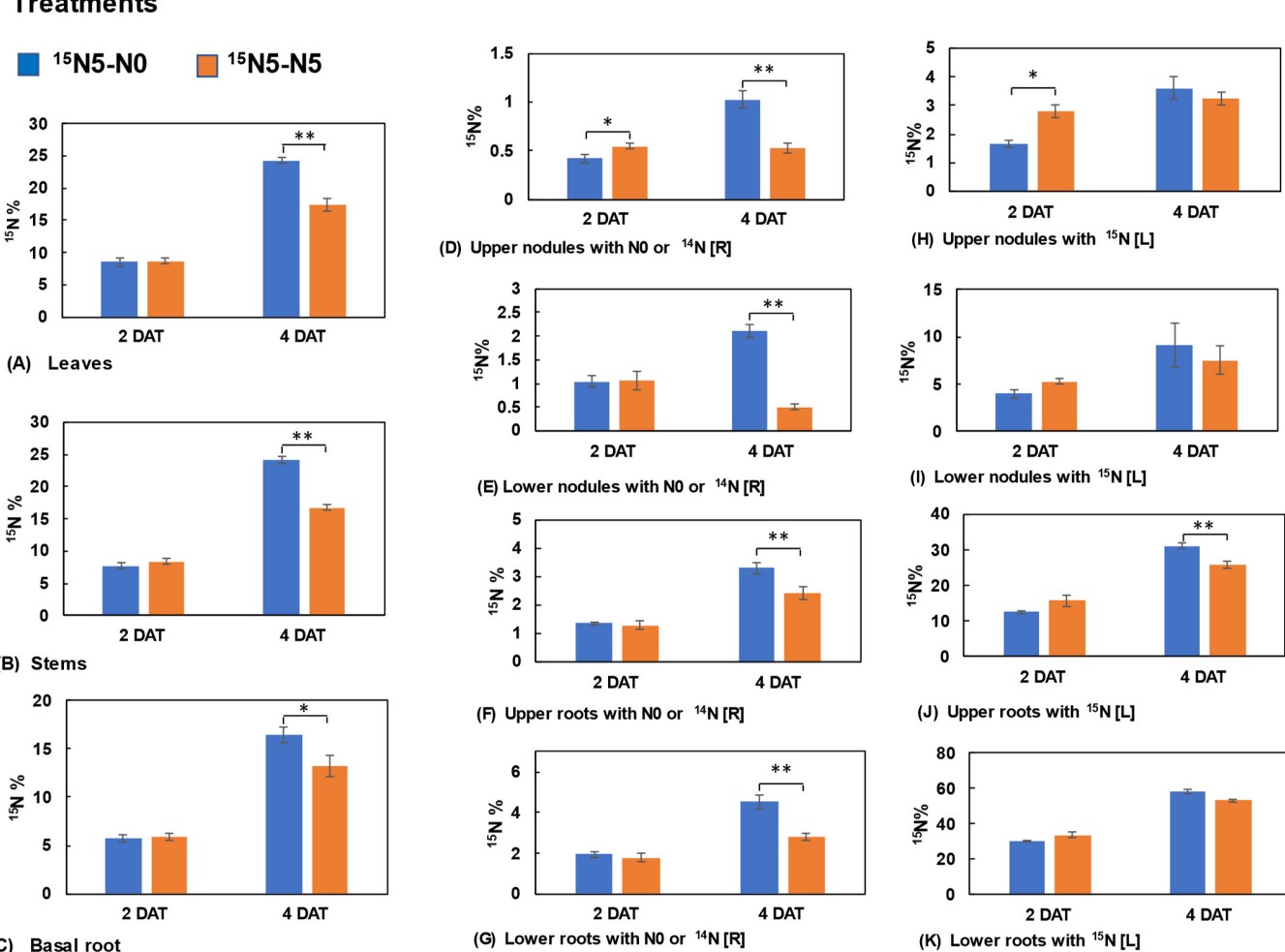

**Figure 4.** Percentage of N derived from $^{15}N$-labeled $^{15}NO_3^-$ in each part of soybean plants with $^{15}N5$-N0 and $^{15}N5$-N5 treatments. (**A**) Leaves, (**B**) Stems, (**C**) Basal roots, (**D**) Upper nodules with N0 or $^{14}N$ [R], (**E**) Lower nodules with N0 or $^{14}N$ [R], (**F**) Upper roots with N0 or $^{14}N$ [R], (**G**) Lower roots with N0 or $^{14}N$ [R], (**H**) Upper nodules with $^{15}N$ [L], (**I**) Lower nodules with $^{15}N$ [L], (**J**) Upper roots with $^{15}N$ [L], (**K**) Lower roots with $^{15}N$ [L]. *, and ** indicate statistical significance between $^{15}N5$-N0 and $^{15}N5$-N5 treatments at $p < 0.05$, and $p < 0.01$ levels based on the Student's T-test. N = (4). Average ± SE.

### 3.1.4. Percentage Distribution of N Derived from $^{15}N$-Labeled Nitrate in Each Part of Soybean Plants

To clarify the effects of the presence or absence of $NO_3^-$ in the opposite side of non-labeled pot on the distribution of $^{15}N$ absorbed from one half-root, the percentage distribution of the $^{15}N$-labeled N in each organ in total $^{15}N$ was compared between $^{15}N5$-N0 and $^{15}N5$-N5 treatments (Table 1). The $^{15}N$ distribution was the highest in leaves at 2 DAT (ca. 46%) and 4 DAT (ca. 56%) and not significantly different between the two treatments (Table 1a). The $^{15}N$ distribution in the stems was relatively constant about 10% irrespective of DAT or treatments. Those in the basal roots were 3% at 2 DAT and 4% at 4 DAT in both treatments.

**Table 1.** Distribution of N derived from $^{15}$N-labeled $NO_3^-$ in each part of soybean plants with split root experiment.

**a. $^{15}$N Distribution in Leaves, Stems, and Basal Roots.**

| Treatment | Days | Leaves | Stems | Basal Roots |
|---|---|---|---|---|
| $^{15}$N5-N0 | 2 DAT | 45.6 (1.1) | 10.2 (0.3) | 2.3 (0.2) |
| $^{15}$N5-N5 | 2 DAT | 47.0 (0.9) | 10.4 (0.6) | 3.7 (0.6) |
| $^{15}$N5-N0 | 4 DAT | 57.8 (0.6) | 10.3 (0.4) | 3.8 (0.1) |
| $^{15}$N5-N5 | 4 DAT | 55.2 (2.3) | 10.8 (1.6) | 4.5 (0.3) |

**b. $^{15}$N Distribution in Nodules and Roots in $^{15}$N-Labeled Pot [L].**

| Treatment | Days | U-Nodules | L-Nodules | U-Roots | L-Roots |
|---|---|---|---|---|---|
| $^{15}$N5-N0 | 2 DAT | 1.04 (0.32) | 3.00 (0.74) | 3.25 (0.38) | 32.0 (0.6) |
| $^{15}$N5-N5 | 2 DAT | 1.07 (0.22) | 1.42 (0.50) | 4.10 (1.54) | 29.6 (1.6) |
| $^{15}$N5-N0 | 4 DAT | 0.73 (0.13) | 0.71 (0.29) | 2.48 (0.41) | 22.1 (1.0) |
| $^{15}$N5-N5 | 4 DAT | 0.75 (0.11) | 0.69 (0.15) | 2.92 (0.28) | 22.7 (1.4) |

**c. $^{15}$N Distribution in Nodules and Roots in Non-Labeled Pot [R].**

| Treatment | Days | U-Nodules | L-Nodules | U-Roots | L-Roots |
|---|---|---|---|---|---|
| $^{15}$N5-N0 | 2 DAT | 0.21 (0.06) | 0.34 (0.06) | 0.35 (0.06) | 1.68 (0.21) |
| $^{15}$N5-N5 | 2 DAT | 0.35 (0.11) | 0.15 (0.07) | 0.37 (0.05) | 1.87 (0.15) |
| $^{15}$N5-N0 | 4 DAT | 0.25 (0.01) | 0.29 (0.06) | 0.29 (0.03) | 1.26 (0.06) |
| $^{15}$N5-N5 | 4 DAT | 0.13 (0.02) | 0.04 (0.02) | 0.32 (0.05) | 1.88 (0.39) |

**Data are means (standard error). N = 4**

U-nodules; upper nodules, L-nodules; lower nodules, U-roots; upper roots, L-roots; lower roots. N = (4). Average ± SE.

Table 1b shows the $^{15}$N distribution of the roots and nodules in a $^{15}NO_3^-$ solution [L]. The $^{15}$N distribution was high at about 31% in the lower roots at 2 DAT, which were in direct contact with the $^{15}NO_3^-$ solution. The $^{15}$N distribution in this part of roots decreased at 4 DAT by about 22%. The $^{15}$N distribution in the lower roots with the $^{15}NO_3^-$ pot was similar between the $^{15}$N5-N0 and $^{15}$N5-N5 treatments. The $^{15}$N distribution of the upper roots in $^{15}NO_3^-$ solution [L] was much lower than the lower roots in the same pot, suggesting that the accumulation of absorbed $^{15}NO_3^-$ is primarily restricted in the root parts in direct contact with $^{15}NO_3^-$ solution. Concerning the $^{15}$N distribution in the lower nodules, the percentages were much lower about 2% at 2 DAT, and 1% at 4 DAT.

The $^{15}$N distribution in the non-labeled pot [R] (Table 1c) indicates that those values were not different between the $^{15}$N5-N0 and $^{15}$N5-N5 treatments. Total distributions in the roots plus nodules in the non-labeled pot were 2.85% and 2.09% at 2 DAT and 4 DAT in $^{15}$N5-N0 treatment and 2.74% and 2.37% at 2 DAT and 4 DAT in $^{15}$N5-N5 treatment. This result suggests that the presence or absence of nitrate in the distal part of the half-roots from the $^{15}NO_3^-$ absorption site did not markedly affect the recycling percentage of $^{15}$N from shoots to the non-labeled half-roots.

### 3.2. Comparisons of Principal N Metabolites in Each Part of the Plants

3.2.1. Nitrate Concentration in Each Part of Soybean Plants

To confirm $NO_3^-$ can be transported from the shoot to the roots and nodules, nitrate concentration in each organ was determined (Figure 5). Nitrate was not detected in the plant parts with N0-N0 treatment where both half-roots were not supplied with $NO_3^-$, although small peaks less than 1 µmol g$^{-1}$ DW were occasionally detected, possibly due to the contamination from the culture solution or environment. When both half-roots were in 5 mM $NO_3^-$ solution ($^{15}$N5-N5), the lower roots markedly accumulated $NO_3^-$ at 2 DAT (452 µmol g$^{-1}$ DW) and 4 DAT (1090 µmol g$^{-1}$ DW) (Figure 5A). The nitrate concentrations in the upper roots were relatively lower than that in the lower roots. The nitrate concentration in the lower nodules that is in direct contact with 5 mM nitrate

solution was not high compared with the lower roots. In the $^{15}$N5-N0 treatment, appreciable concentration of nitrate was not detected in all parts of the N-free [R] pot, i.e., upper roots, lower roots, upper nodules, and lower nodules (Figure 5B). This result indicates that the absorbed $NO_3^-$ in one side of the half-roots is not translocated to the other side as it is. The lower part of the roots in the [L] pot with $^{15}$N5-N0 accumulated $NO_3^-$ at 2 DAT (547 μmol g$^{-1}$ DW) and 4 DAT (927 μmol g$^{-1}$ DW) (Figure 5B), and these values were similar to the lower roots with $^{15}$N5-N5 treatment (Figure 5A). Nitrate concentrations in the upper roots, and upper and lower nodules in [L] pot with $^{15}$N5-N0, were almost the same as in the corresponding parts in the $^{15}$N5-N5 treatment. In addition, nitrate concentrations in leaves, stems, and basal roots were not different between the $^{15}$N5-N0 treatment and the $^{15}$N5-N5 treatment.

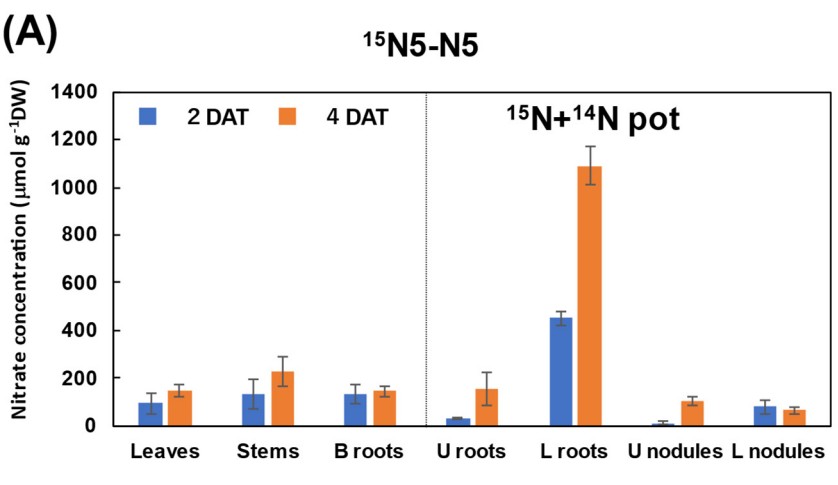

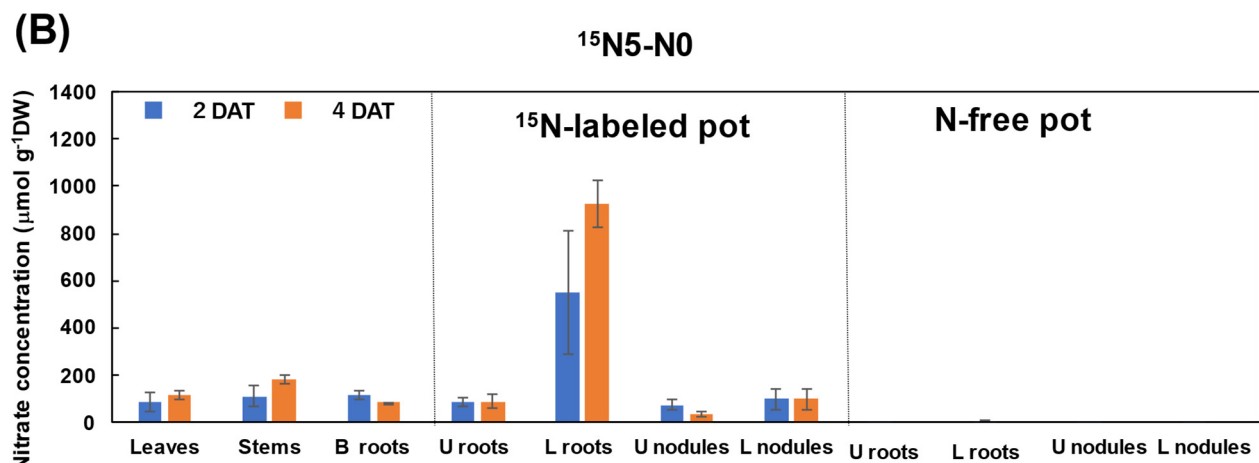

**Figure 5.** Nitrate concentration in each part of soybean with split-root treatments. (**A**) $^{15}$N5-N5 treatment: The data of the roots and nodules in $^{15}NO_3^-$ and $NO_3^-$ pots are shown. (**B**) $^{15}$N5-N0 treatment: The data of the roots and nodules in $^{15}NO_3^-$ and those in N-free pots are separately shown. N = (4) (**A**) leaves, stems, and basal roots, and (**B**) roots and nodules, N = (8) (**A**) roots and nodules. Average ± SE.

### 3.2.2. N Concentrations of Major N Metabolites in Leaves, Stems, and Basal Roots

The concentrations of Glu, Asp, Asn, and allantoate in the leaves tended to be high with $^{15}$N5-N5 > $^{15}$N5-N0 > N0-N0 treatment, although the concentrations of Gln and allantoin were higher in N0-N0 treatment at 4 DAT than the other treatments (Figure 6A,D). The Asn concentration in the stems was significantly higher in the $^{15}$N5-N5 than N0-N0 treatment at 4 DAT (Figure 6B,E). In the basal roots, the concentrations of Glu, Asp, Gln, and Asn were the highest in $^{15}$N5-N5, compared with the $^{15}$N5-N0, and N0-N0 treatments

at 4 DAT (Figure 6C,F). The concentrations of allantoin and allantoate were not different among treatments at 4 DAT, although these were higher in the $^{15}$N5-N5 treatment at 2 DAT.

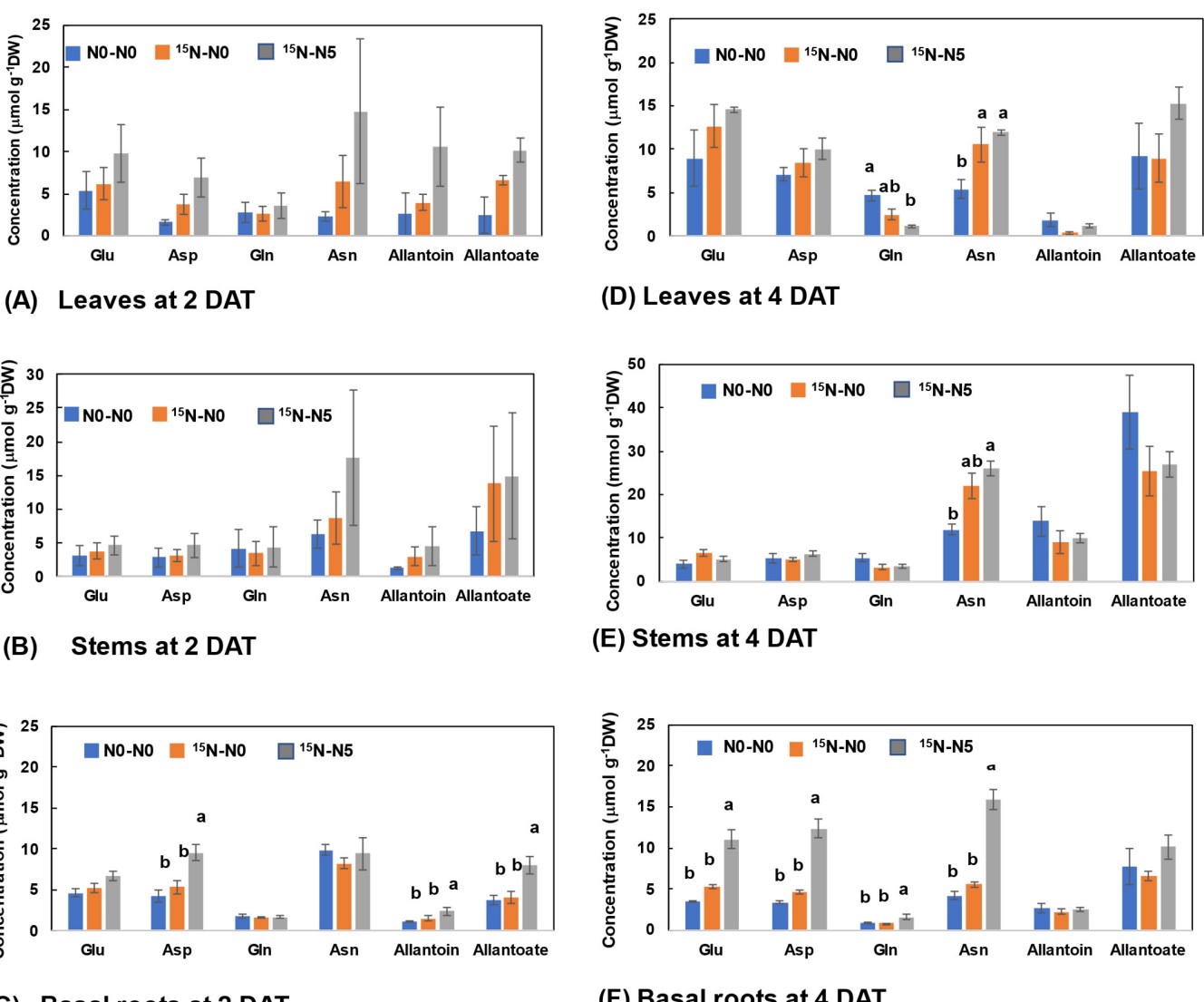

**Figure 6.** Comparison of the concentrations of principal N metabolites in the leaves, stems, and basal roots of soybean plants with split root experiments with N0-N0, $^{15}$N5-N0, and $^{15}$N5-N5 treatments. (**A**) Leaves at 2 DAT, (**B**) Stems at 2 DAT, (**C**) Basal roots at 2 DAT, (**D**) Leaves at 4 DAT, (**E**) Stems at 4 DAT, (**F**) Basal roots at 4 DAT. The different letters indicate significant differences among treatments based on Tukey's test ($p < 0.05$). Different letters indicate a significant difference ($p < 0.5$) based on Tukey's test. N = (4). Average ± SE.

### 3.2.3. N Concentrations of Principal N Metabolites in the Upper and Lower Roots

The concentrations of N metabolites were compared between roots and nodules of the plants with N0-N0 treatment and $^{15}$N5-N5 treatment (Figures 7A,B and 8A,B). In addition, these were compared in the single plant where one half-root was supplied with 5 mM $^{15}$NO$_3^-$ and another half was supplied N-free solution (Figures 7C,D and 8C,D). Figure 7A,B shows the concentrations of major N metabolites in the upper roots and the lower roots comparing N0-N0 treatment which depends on only N$_2$ fixation, and $^{15}$N5-N5 treatment in which both sides of half-roots were supplied with 5 mM NO$_3^-$. The concentrations of Gln, Asp, and Asn significantly increased by supplying NO$_3^-$ for 2 days, while the concentrations of allantoin and allantoate decreased at 4 DAT.

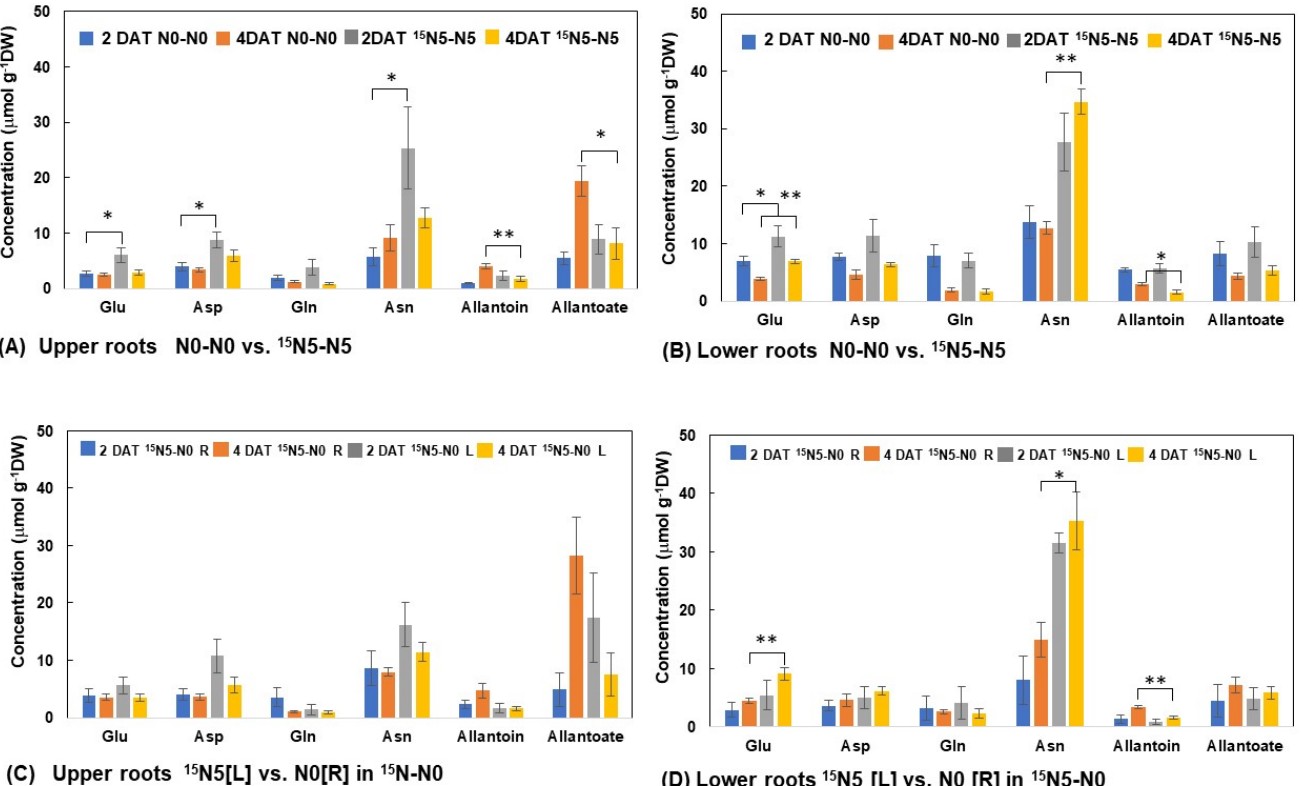

**Figure 7.** Comparison of the concentrations of principal N metabolites in the upper roots (**A**) and the lower roots (**B**) of soybean plants with split root experiments with N0-N0 and $^{15}$N5-N5 treatments, and the concentrations of N compounds in the upper roots (**C**), and the lower roots (**D**) of soybean in which either half-root were supplied with N-free solution of 5 mM nitrate in $^{15}$N5-N0 treatment. The * and ** indicate the significant differences between N0 and N5 conditions based on Student's T-test ($p < 0.01$, and $0.01 < p < 0.05$). (**A**) N = (8), (**B**) N = (4). Average ± SE.

Figure 7C,D shows the concentrations of major N metabolites on each side of the half-roots of the plants treated with $^{15}$N5-N0 treatment. The $^{15}$NO$_3^-$ was supplied in the solution in [L] pot, and the N-free solution in [R] pot. The trends in the concentrations by 5 mM NO$_3^-$ supply were similar to the upper (Figure 7A) and lower (Figure 7B) roots of both half-roots supplied with either N-free solution or the solution with 5 mM NO$_3^-$. The concentrations of Glu and Asn significantly increased in the lower roots supplied with 5 mM NO$_3^-$ compared with N-free solution, whereas the concentration of allantoin decreased at 4 DAT. These results indicate that the concentrations of major N metabolites directly depended on the conditions surrounding the half-roots and were not significantly affected by the opposite half-roots whether N-free or 5 mM NO$_3^-$.

3.2.4. N Concentrations of Principal N Compounds in the Upper and Lower Nodules

Figure 8A,B shows the concentrations of principal N compounds in the upper nodules and the lower nodules comparing N0-N0 and $^{15}$N5-N5 treatments. The concentration of Glu in the nodules was 20–40 μmol g$^{-1}$ DW and higher than those in the roots, which is less than 10 μmol g$^{-1}$ DW, and the concentration was not affected by the treatments. The concentration of Asp tended to increase in the nodules by supplying NO$_3^-$, whereas the concentrations of allantoate decreased by NO$_3^-$ supply.

Figure 8C,D shows the concentrations of principal N compounds in the upper and lower nodules attached to each side of the half-roots of a single plant treated with $^{15}$N5-N0 treatment. The concentrations and the trends by 5 mM NO$_3^-$ supply were similar to the upper (Figure 8A) and lower (Figure 8B) nodules with N0-N0 and $^{15}$N5-N5 treatments, and the increase in Asn concentration at 4 DAT was remarkable only in the half-roots with

$^{15}NO_3^-$. The concentration of allantoate tended to decrease at 4 DAT in the half-roots with 5 mM $NO_3^-$ solution. These results indicate that the concentrations of principal N compounds in nodules were not affected compared with those in the roots.

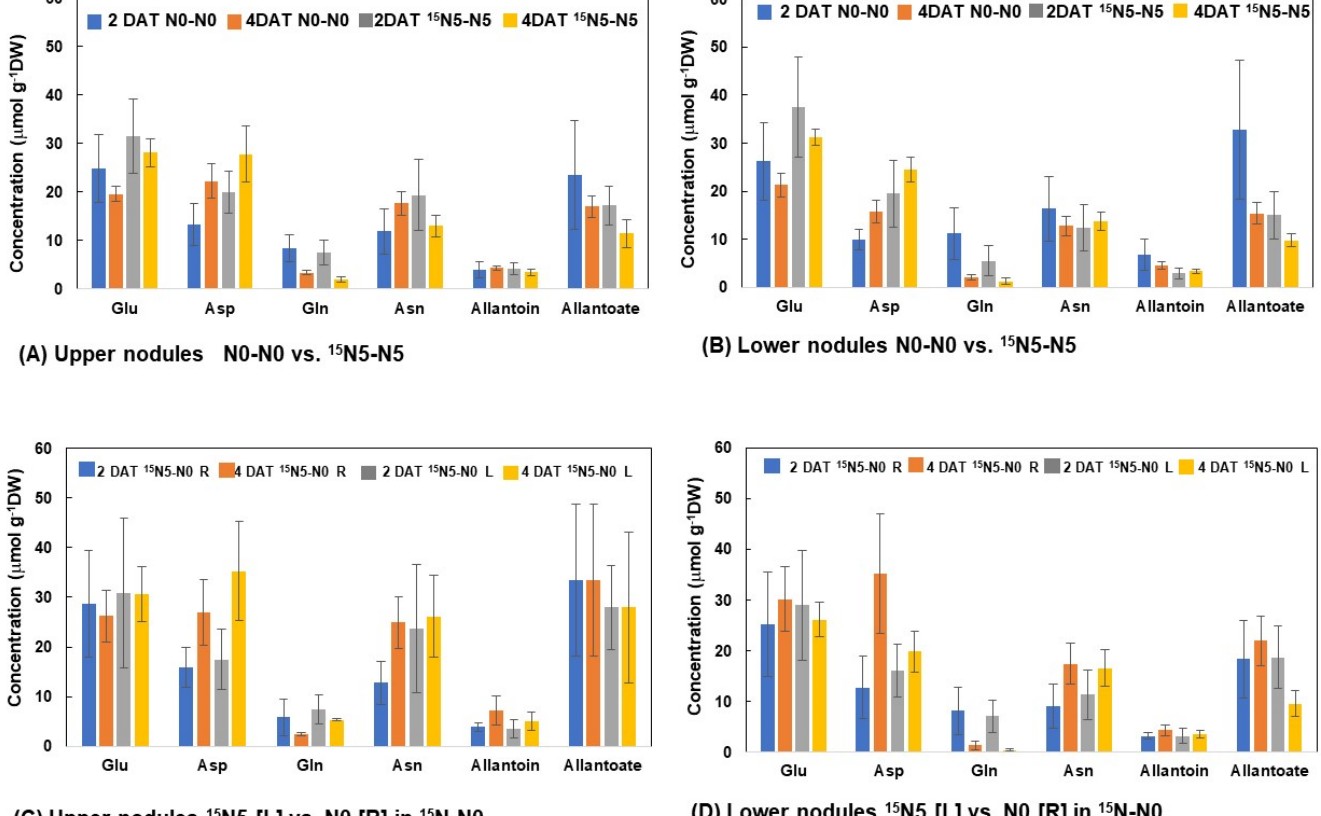

**Figure 8.** Comparison of the concentrations of principal N metabolites in the upper nodules and the lower nodules of soybean plants with split root experiments with N0-N0 and $^{15}$N5-N5 treatments and the concentrations of N compounds in either half-roots supplied with N-free solution or 5 mM nitrate in $^{15}$N5-N0 treatment. (**A**) Upper nodules, N0-N0 vs. $^{15}$N5-N5, (**B**) Lower nodules, N0-N0 vs. $^{15}$N5-N5, (**C**) Upper nodules, $^{15}$N-N0, $^{15}$N5(L) vs. N0 (R), (**D**) Lower nodules, $^{15}$N-N0, $^{15}$N5(L) vs. N0 (R). (**A**) N = (8), (**B**) N = (4). Average ± SE.

## 4. Discussion

### 4.1. Changes in Total N and N Derived from $^{15}$N-Labeled Nitrate with Split-Root Systems

Soybean plants utilize the N fixed by root nodules and the N absorbed from the roots. However, the nodulation, nodule growth, and $N_2$ fixation activity are inhibited when the nodulated roots are in direct contact with high concentrations of combined N, especially nitrate [13–17]. The effects of nitrate on nodule growth vary by nitrate concentrations, treatment period, placement in the medium [18,19], and legume species [20,21]. In addition, local and systemic effects of $NO_3^-$ are distinguished; local (direct) effects where nitrate is supplied directly to the nodulated roots and systemic (indirect) effects where nitrate applied from the distal part of the roots [18,19]. Fijikake et al. [12,22] found that the direct effect of 5 mM nitrate on the soybean nodule growth and nitrogen fixation activity was quick and reversible in soybean plants. Nodule growth was suppressed at several hours after addition of 5 mM of nitrate [23,24]. Transcriptome and metabolome analysis after one day of 5 mM $NO_3^-$ treatment supported that nitrate promotes nitrogen and carbon metabolism in the roots but repressed it in the nodules [25]. The quick and reversible inhibition of nodule growth and nitrogen fixation activity was also observed by ammonium, urea, and glutamine, although the inhibitory effects were less than nitrate [26].

In this experiment, the total amounts of N were not significantly different among N0-N0, $^{15}$N5-N0, and $^{15}$N5-N5 treatments at 2 DAT and 4 DAT, indicating that the amount of N from absorbed $NO_3^-$ was almost equivalent to the decrease in the fixed N due to repression of nitrogen fixation by $NO_3^-$ under this experiment conditions. Based on the total N accumulation and $^{15}$N absorption from 2 DAT to 4 DAT, the percentage dependences on Ndfa and Ndfn were estimated. The %Ndfa decreased to 59% and 46% when one half-root or both half-roots supplied with $NO_3^-$. The amount of $^{15}$N absorption was higher in the second period of 2 DAT to 4 DAT than the first period from 0 DAT to 2 DAT in both $^{15}$N-N0 and $^{15}$N-N5 treatments. These results might be due to the induction period of nitrate transporters in the roots or the accelerating growth of the plants. In all treatments, the increase of N in leaves from 2 DAT to 4 DAT was remarkable for about 3/4 of the total N increase, because the leaf growth was vigorous in this stage (about V4). In addition, N fixation activity in this stage might become increasing.

The 40% higher $^{15}$N absorption in the second 2 days from 2 DAT to 4 DAT with $^{15}$N5-N0 than that with $^{15}$N5-N5 (Figure 3) and the higher $^{15}$N% in the roots and nodules in [L] pot with N-free solution (Figure 4) might be due to the compensation of nitrate absorption due to only half-roots absorbing $NO_3^-$. Similar upregulation of $NO_3^-$ absorption was reported in *Brassica napus* L. using split-root systems, and the uptake rates by half-roots with $NO_3^-$ increased by about two-fold when the opposite side of half-root was in N-free solution within 24 h [27]. Forde [2] suggested that $NO_3^-$-fed plants can compensate for a local N limitation and maintain their N status by stimulating $NO_3^-$ acquisition by roots exposed to mineral N. A short-term adaptation response may be mediated by a rapid induction of the nitrate uptake systems in these roots. High-affinity nitrate transporters involved in these responses were identified in Arabidopsis [28,29]. From the results by Laine [27], the nitrate absorption rates correlated with total N and $NO_3^-$ concentrations in the shoot, but not with concentrations of each amino acid. The author suggested that total N and $NO_3^-$ concentrations in the shoot may regulate the nitrate absorption rate in Brassica. Alternatively, stimulation of the $NO_3^-$ absorption in the $NO_3^-$ supplied half-roots with N-free solution in the opposite half-roots might be related to the promotion of the photoassimilate supply to the half-roots with 5 mM $NO_3^-$ compared with the N-free solution [12]. When plant leaves were exposed to $^{11}$C-labeled $CO_2$ to the soybean plants with split-root systems, of which one half-root was supplied with 5 mM $NO_3^-$ and the opposite half-roots received an N-free solution, the translocation of $^{11}$C labeled-C in the nitrate-fed half-root was stimulated compared with N-free half-roots [12].

Although $^{15}$N absorption by the half-roots in the $^{15}NO_3^-$ pot [L] was accelerated by N-free conditions in the opposite pot [R], the $^{15}$N distribution in the non-labeled pot [R] (Table 1c) was not significantly affected by the N conditions in the [R] pot, either N-free or 5 mM $NO_3^-$. The sum of $^{15}$N distributions in the roots plus nodules in the non-labeled pot were 2.85% and 2.09% at 2 DAT and 4 DAT in $^{15}$N5-N0 treatment, and 2.74% and 2.37% at 2 DAT and 4 DAT in $^{15}$N5-N5 treatment. In addition, the percentage distribution of $^{15}$N in leaves, stems, basal roots, and the roots and nodules of the $^{15}$N absorption site [L] were similar between $^{15}$N5-N0 and $^{15}$N5-N5 treatments. The results indicate that the different N conditions with $+^{15}$N$-$N and $+^{15}$N$+$N did not change the transport and recycling of $^{15}$N absorbed in the $^{15}$N absorbing half-root [L]. A similar result was reported, and the recycling of $^{15}$N supplied from one half-root of soybean Williams, and its hypernodulation mutant lines NOD1-3, NOD2-4, and NOD3-7 after 2 days of $^{15}NO_3^-$ feedings, were about 1.5–1.7% of total $^{15}$N, and not different among Williams and the hypernodulation lines [30]. This indicates that the N recycling itself may not be involved in the autoregulation of nodulation.

### 4.2. Transport Pathways of Nitrate Absorbed in the Half-Roots Supplied with $^{15}NO_3^-$ to the Shoot and the Opposite Side of Roots

The concentrations of $NO_3^-$ in the lower roots in direct contact with 5 mM $NO_3^-$ were the highest among plant parts after 2 and 4 DAT, and the $NO_3^-$ concentrations were not affected by the opposite side of half-roots with N-free solution or non-labeled $NO_3^-$.

This result indicated that nitrate accumulation in the lower roots is independent of the N conditions of the other half-roots. The lower nodules, which are also in direct contact with 5 mM $NO_3^-$ accumulated less than 1/5 of the lower roots (Figure 5A,B). The $NO_3^-$ absorption in nodules was reported through a nodule surface, and not transported via the xylem or phloem in the roots [31].

The $NO_3^-$ accumulation in the basal roots, stems, and leaves was observed even at 2 DAT, but $NO_3^-$ was not detected in the other half-roots with N-free pot in $^{15}$N5-N0 as same as N0-N0 treatment. This result clearly shows that $NO_3^-$ itself did not readily recycle from leaves to the roots. Li et al. [32] reported that $NO_3^-$ absorbed from the N-supply side of the roots of the dual-root system of soybean might be transported via the basal root pealed skin (via phloem) and woody part of the roots (via xylem). The discrepancy between our results and theirs may be the split-root system by cutting a primary root or the dual-root system made by grafting two seedlings. Most of the recycled N from the shoot may be transported through the phloem. [33].

### 4.3. Changes in Amides, Amino Acids, and Ureides in Each Part of the Plants with Split-Root Systems

Phloem-borne amino acids have been prominent candidates for the role of shoot-derived signal for feedback regulation of $NO_3$ uptake [34,35]. Rapid cycling of amino acids between the shoot and the root may occur, so the changes in the N status in the shoot will rapidly reflect the delivery rate of amino acids to the roots [36,37]. However, contrary to the requirements of this model, N deficiency can sometimes lead to an increase rather than a decline in amino acid cycling [38]. Furthermore, split-root studies on mung bean (*Ricinus communis*) found no correlation between $NO_3^-$ uptake rate and the amino acid content of the phloem [39].

There are many hypotheses for the causes of nitrate inhibition of nodulation and nitrogen fixation, carbohydrate deprivation in nodules, feedback inhibition by a product of nitrate metabolism such as asparagine or ureides (allantoate and allantoin) [31,32,40,41], and the decrease in oxygen diffusion into the nodules which restricts the respiration of bacteroid, a symbiotic state of rhizobia in nodules [36,42–44]. The concentrations of Asn in the lower roots supplied with 5 mM nitrate at 4 DAT were significantly higher at 35 μmol g$^{-1}$ DW in $^{15}$N5-N5 treatment than those in N0-N0 treatment about 13 μmol g$^{-1}$ DW (Figure 6B). The same was true for the half-roots with $^{15}$N5 [L] pot at 35 μmol g$^{-1}$ DW and N0 [R] pot at 15 μmol g$^{-1}$ DW (Figure 6D). These results indicate that some part of nitrate absorbed by the half-roots in direct contact with 5 mM $NO_3^-$ was assimilated there, and the metabolite Asn accumulated. The stimulation of Asn accumulation in the opposite side of N-free roots was not observed, so the acceleration of Asn transport from the $^{15}NO_3^-$-fed half-roots via shoot to the N-free roots did not occur in these experimental conditions. The changes in the concentrations of N compounds in nodules were not apparent either in the +N or −N site (Figure 7), suggesting that $NO_3^-$ absorption by root nodules from culture solution is much less than that from the roots.

### 5. Conclusions

Characteristics of absorption, transport, and recycling of nitrogen investigated using split-root systems of nodulated soybean plants, supplying $^{15}NO_3^-$ to one half-root, and another half-root with N-free solution or non-labeled nitrate. A high accumulation of nitrate was detected in the roots part in direct contact with 5 mM $NO_3^-$, but nitrate was absent in the roots and nodules in the N-free pot, suggesting that nitrate itself did not recycle as it was from the shoot to the roots. The $^{15}NO_3^-$ absorption in the half-roots with N-free solutions in opposite half-roots was 40% accelerated from 2 to 4 days of nitrate treatment, compared with those with non-labeled $NO_3^-$. The recycling percentages of $^{15}$N in the roots plus nodules in the opposite N-free and non-labeled $NO_3^-$ half-root were the same at 2–3%. The concentrations of amino acids, especially Asn, increased in the

half-roots in direct contact with $NO_3^-$ as well as basal roots, stems, and leaves but Asn did not accumulate in the opposite half-roots with N-free solution.

**Author Contributions:** Conceptualization, T.O.; methodology, T.O., A.S., T.S. and T.S.; software, T.O., T.S. and K.H.; validation, T.O., A.S. and K.H.; formal analysis, M.D.; investigation, T.O.; data curation, T.O.; writing—original draft preparation, T.O.; writing—review and editing, T.O.; visualization, T.O.; supervision, T.O.; project administration, T.O. All authors have read and agreed to the published version of the manuscript.

**Funding:** This research received no external funding.

**Data Availability Statement:** The data presented in this study are available on request from the corresponding author.

**Conflicts of Interest:** The authors declare no conflict of interest.

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
