# Peer review of "N Absorption, Transport, and Recycling in Nodulated Soybean Plants by Split-Root Experiment Using 15N-Labeled Nitrate"

_nitrogen, doi:10.3390/nitrogen3040042_

Round 1

Reviewer 1 Report

The experimental design in the manuscript is reasonable and the results are detailed. There are some small problems that need to be modified and released after consideration. 

N5-N0 2 in Figure 2, N5-N5 2 not consistent with Figure 1 and ' 2 DAT ' not aligned with Figure 2 in Figure 1.  The conclusion is too long, just a list of results, no longer concise.

1. The manuscript is not very innovative, the use of split root soybean 15N marker test, has been found in previous studies, but the manuscript data more adequate.

2. The title of the manuscript is about recycling of nitrogen, and the discussion should include a picture detailing the flow of nitrogen from roots and nodules.

3. Line 529-545 describes the transport process of hormones and other substances, but it has no obvious relationship with this study. The last sentence reflects the theme, which is related to N transport.

4. Discussion 4.4 part, the relative expression of gene changes can be written together with the corresponding indicators, alone to discuss the gene has no specific meaning

5. There is no footnote in table 1. It is not clear what the two numbers represent.

6. Discussion in the manuscript is difficult to support the title, and there are some outcome data that are not reflected, such as indicators such as sucrose synthase gene expression.

7. The statements in the abstract and conclusion sections still describe the result analysis and do not highlight the conclusions of the manuscript.

Author Response

Dear Reviewer 1

Thank you very much for reviewing our manuscript and gave us valuable comments.

Comments and Suggestions for Authors

The experimental design in the manuscript is reasonable and the results are detailed. There are some small problems that need to be modified and released after consideration. 

Ans: Thank you very much for kind comments.

N5-N0 2 in Figure 2, N5-N5 2 not consistent with Figure 1 and ' 2 DAT ' not aligned with Figure 2 in Figure 1.  The conclusion is too long, just a list of results, no longer concise.

Ans: I am sorry for mistake in Figure 2. I revised the Figures carefully.

I revised the conclusions by your advise.

  1. The manuscript is not very innovative, the use of split root soybean 15N marker test, has been found in previous studies, but the manuscript data more adequate.

Ans. Thank you very much for your comments.

  1. The title of the manuscript is about recycling of nitrogen, and the discussion should include a picture detailing the flow of nitrogen from roots and nodules.

Ans: I revised the title to “N absorption, transport, and recycling in nodulated soybean plants by split-root experiment using 15N-labeled nitrate”. I added a graphic abstract for the flow of nitrogen from roots and nodules in this experiment.

  1. Line 529-545 describes the transport process of hormones and other substances, but it has no obvious relationship with this study. The last sentence reflects the theme, which is related to N transport.

Ans: I deleted the line 529-545.

  1. Discussion 4.4 part, the relative expression of gene changes can be written together with the corresponding indicators, alone to discuss the gene has no specific meaning

Ans: Reviewer 3 pointed out that quantifying bands on a gel to measure gene expression is outdated methodology. Transcript levels of target genes should be measured by Real Time-qPCR and normalized against at least two housekeeping/reference genes. So, I deleted all the gene expression data from manuscript.

  1. There is no footnote in table 1. It is not clear what the two numbers represent.

Ans. I added the foot note in Table 1.

  1. Discussion in the manuscript is difficult to support the title, and there are some outcome data that are not reflected, such as indicators such as sucrose synthase gene expression.

Ans. I revised the discussion part.

  1. The statements in the abstract and conclusion sections still describe the result analysis and do not highlight the conclusions of the manuscript.

Ans. I revised the abstract and conclusion.

Thank you very much.

Takuji Ohyama

Reviewer 2 Report

The manuscript by Doi et al. delivers few interesting accounts on nitrogen uptake and recycling in soybean plant under heterogenous supply. Experiments are conducted with care and provide a large amount of data. Unfortunately, the manuscript is poorly composed. The authors submitted a draft version that requires a substantial editorial effort and a clearer presentation of the figures. The text is dense and the syntax needs to be reworked. The train of thoughts is really hard to follow. Scientific highlights are not clearly depicted in the abstract and in the conclusion. Which is the novel message of the paper?

The introduction is over lengthy. It should go straight to the point and finish by drawing research questions.

The materials and methods section should be shortened. Some paragraphs depicting plant culture are duplicated between the different sections.

Ln 116-117: At which distance from the hypocotyl is the primary root cut?

Ln 120-121: Rephrase à under a photoperiod of 16 h (228 µmol photons m-2 s-1)/8 h darkness

Ln 175-176: Gene names should be written in capital and italic (check for exact nomenclature, this has to be consistent through the manuscript). First, gene names appear in full, then they can be abbreviated.

Ln 185: sec à s

Table S1 (the list of primers for PCR) is not referred in the body text and not listed in the supplementary materials.

Quantifying bands on a gel to measure gene expression is outdated methodology. Transcript levels of target genes should be measured by Real Time-qPCR and normalized against at least two housekeeping/reference genes.

The results section starts too abruptly. A short introductory paragraph may help guiding the reader. The whole narrative of the results must be redrafted.

The discussion is largely repeating results.

Figures

Fig. 1 Apparatus for a split-root experiment à Split-root experimental setting

Indicating the different root portions (lower/upper right/left nodules or roots) would help the reader to better visualize the experimental setting.

For all the figures in the manuscript, the graduations on the Y axes are missing.

The visibility of Figures 2 and 3 needs to be improved. They are too many colors and this makes treatment comparison really difficult. These two figures share the same title and legend. The number of observations should be indicated ± std.

All panels in Fig. 4 and Fig. 7 should be described in the legend.

Fig. 9 is puzzling.

Author Response

Dear Reviewer 3

Thank you very much for reviewing our manuscript and giving us valuable comments.

Comments and Suggestions for Authors

The manuscript by Doi et al. delivers few interesting accounts on nitrogen uptake and recycling in soybean plant under heterogenous supply. Experiments are conducted with care and provide a large amount of data. Unfortunately, the manuscript is poorly composed. The authors submitted a draft version that requires a substantial editorial effort and a clearer presentation of the figures. The text is dense and the syntax needs to be reworked. The train of thoughts is really hard to follow. Scientific highlights are not clearly depicted in the abstract and in the conclusion. Which is the novel message of the paper?

Ans: I am sorry that the first manuscript was confusing and not clear. I revised the figures and text.

The introduction is over lengthy. It should go straight to the point and finish by drawing research questions.

Ans: I revised and shorten the introduction part as your suggestions.

The materials and methods section should be shortened. Some paragraphs depicting plant culture are duplicated between the different sections.

Ans: I deleted the gene expression parts, and the materials and methods section become shortened.

Ln 116-117: At which distance from the hypocotyl is the primary root cut?

Ans: I cut off the primary root at about 5 cm in length. I remained the lateral roots.

Ln 120-121: Rephrase  under a photoperiod of 16 h (228 µmol photons m-2 s-1)/8 h darkness

Ans: I am sorry for the rephrasing. I revised this part.

Ln 175-176: Gene names should be written in capital and italic (check for exact nomenclature, this has to be consistent through the manuscript). First, gene names appear in full, then they can be abbreviated.

Ans: I deleted the gene expression part.

Ln 185: sec  s

Table S1 (the list of primers for PCR) is not referred in the body text and not listed in the supplementary materials.

Ans: I deleted the gene expression part.

Quantifying bands on a gel to measure gene expression is outdated methodology. Transcript levels of target genes should be measured by Real Time-qPCR and normalized against at least two housekeeping/reference genes.

Ans: Thank you for your comments. We have a Real Time qPCR apparatus now, but we did not have this apparatus when Ms. Doi analyzed the expression of N related genes in her MS course study. So we decided to delete all the gene expression data from the manuscript. The manuscript may be more concise and clear without gene expression data.

The results section starts too abruptly. A short introductory paragraph may help guiding the reader. The whole narrative of the results must be redrafted.

Ans: I am sorry for the result section. I added some sentence at the beginning of the paragraph.

The discussion is largely repeating results.

Ans: I revised the discussion part.

Figures

Fig. 1 Apparatus for a split-root experiment  Split-root experimental setting

Indicating the different root portions (lower/upper right/left nodules or roots) would help the reader to better visualize the experimental setting.

Ans: Thank you very much for your advise. I revised as your comments.

For all the figures in the manuscript, the graduations on the Y axes are missing.

Ans. I am sorry to delete the Y axes in the figures. I added them in all the figures.

The visibility of Figures 2 and 3 needs to be improved. They are too many colors and this makes treatment comparison really difficult. These two figures share the same title and legend. The number of observations should be indicated ± std.

Ans: I changed the placement of Figure legend at the right side.

All panels in Fig. 4 and Fig. 7 should be described in the legend.

Ans: I added all panels in Figs 4 and 7 in the legends.

Fig. 9 is puzzling.

Ans: Fig 9 was deleted.

Best regards

Takuji Ohyama

Reviewer 3 Report

Comments to the authors:

1. The description of the results can be improved to convey a clear message and not confusing the readers.

2. The authors should harmonize the display of figures in the manuscript (See Figures 4 to 8) for clarity and easy understanding.

3. Discussion, lines 518-545: the authors should critically discuss how the expression patterns of tested genes could explain the recorded nitrate use and assimilation.

4. Page 25-27: qPCR results has two figures without indication of figure number, are they supplementary figures? I yes, please indicate and cite them in the text accordingly.

The authors should also write on the y-axis "relative expression fo "gene name".

I recommend the authors to modify the display of data in the qPCR results. For instance, the panel showing the expression of NT1 gene in leaf samples (left side) can be placed next (right side of the same row) to NT1 in roots followed by that in nodules to facilitate the reading and understanding of data. You may do this for all the target genes.

The resolution of the figures is not good for upper panels. The authors should assign panels number or letters for easy citation and clarity

6. The conclusion can be improved to deliver a take-hope message highlighting with clarity the findings of this study

Author Response

Dear Reviewer 2

Thank you very much for reviewing our manuscript and gave us valuable comments and suggestions. I am sorry that there are many uncleared sentences.

Comments and Suggestions for Authors

Comments to the authors:

  1. The description of the results can be improved to convey a clear message and not confusing the readers.

Ans: I am sorry for the results were not clear and confusing the readers. I revised the results extensively.

  1. The authors should harmonize the display of figures in the manuscript (See Figures 4 to 8) for clarity and easy understanding.

Ans: I revised the Figures.

  1. Discussion, lines 518-545: the authors should critically discuss how the expression patterns of tested genes could explain the recorded nitrate use and assimilation.

Ans: Reviewer 3 pointed out that quantifying bands on a gel to measure gene expression is outdated methodology. Transcript levels of target genes should be measured by Real Time-qPCR and normalized against at least two housekeeping/reference genes. So, I deleted all the gene expression data from the manuscript.

  1. Page 25-27: qPCR results has two figures without indication of figure number, are they supplementary figures? I yes, please indicate and cite them in the text accordingly.

Ans: I deleted this part.

  1. The authors should also write on the y-axis "relative expression fo "gene name".

I recommend the authors to modify the display of data in the qPCR results. For instance, the panel showing the expression of NT1 gene in leaf samples (left side) can be placed next (right side of the same row) to NT1 in roots followed by that in nodules to facilitate the reading and understanding of data. You may do this for all the target genes.

The resolution of the figures is not good for upper panels. The authors should assign panels number or letters for easy citation and clarity

Ans: Thank you for the comments. I missed the y-axis for all figures. I added it.

  1. The conclusion can be improved to deliver a take-hope message highlighting with clarity the findings of this study

Ans: Thank you for your comment. I revised the conclusion part.

Takuji Ohyama

Round 2

Reviewer 2 Report

The authors provided a quick revision of the manuscript. They decided to remove the gene expression survey. The text was indeed shortened but substantial editorial effort still needs to be done. The English language and style need to be reworked extensively. Some sentences are difficult to understand.

It is not possible to cover every item but these are three examples showing how the text can be improved:

My previous comment was not addressed. Figures 2 and 3 share the same title. The number of observations should be indicated ± std.

Nitrogen content in each part of soybean plants with N0-N0, 15N5-N0, and 15N5-14N5 treatments at 2 and 4 DAT. Average and standard error. Different alphabet on the top of the columns indicates the significantly different in total N among treatments based on Tukey’s Test (P<0.05). Different alphabet at the side of the column indicates the significant difference of N content in the parts among treatments. The alphabet was omitted in the part with no significant difference.

My suggestion:

Nitrogen content in soybean plants treated with 15N label after 2 and 4 days of treatment. The experimental setting is described in the legend of Fig. 1. Different letters indicate significant difference (P<0.5) based on a Tukey’s test. N = (give number of observations) ± SE.

 Ln 30-36

Plants absorb N from their roots primarily in the forms of nitrate and ammonium in soils or from fertilizers when applied [1]. Leguminous plants such as soybean can also utilize atmospheric N2 fixed in the root nodule, a symbiotic organ with soil bacteria rhizobia. The nitrate absorbed in the soybean roots is translocated to the shoot in the form of NO3- or it is assimilated in the roots and transported to the leaves mainly in the form of asparagine (Asn) via xylem vessels.

My suggestion:

Plant roots absorb N primarily in the forms of nitrate (NO3-) and ammonium (NH4+). Leguminous plants like soybean, incorporate atmospheric dinitrogen (N2) in root nodules, forming a symbiosis with N2-fixing rhizobia. Nitrate absorbed by soybean roots can directly be translocated to the shoot, or first assimilated and then transported mainly as asparagine (Asn), via xylem vessels to the leaves.

 Ln 97-99

Two 1000 mL polyethylene terephthalate pots were bound, and each pot was independently filled with 800 mL of N-free culture solution covered with aluminum foil with continuous aeration.

My suggestion:

Two polyethylene terephthalate pots with a capacity of 1 L were attached. Each pot was filled with 0.8 L of N-free culture solution, covered with aluminum foil and aerated continuously.

 Ln 44 : glutamine (Gln)

Ln 68: glutamate (Glu), aspartate (Asp)

Ln 85-86 My previous comment was not addressed.

under a photon flux density of 228 mmol m-2 s-1 with a 16-hour photoperiod and 8-hour dark period.

The light unit is wrong.

under a photoperiod of 16 h light (228 µmol photons m-2 s-1)/ 8 h darkness

Ln 121: N metabolites instead of N constituents

Author Response

Answers for Comments and Suggestions by reviewer 2

The authors provided a quick revision of the manuscript. They decided to remove the gene expression survey. The text was indeed shortened but substantial editorial effort still needs to be done. The English language and style need to be reworked extensively. Some sentences are difficult to understand.

Answer: Thank you very much for valuable comments and suggestions. I am sorry that there are some sentences difficult to understand. I revised the manuscript.

It is not possible to cover every item but these are three examples showing how the text can be improved:

Answer: Thank you very much for your kind suggestions to show how the text can be improved. I revised following your advise.

My previous comment was not addressed. Figures 2 and 3 share the same title. The number of observations should be indicated ± std.

Answer: I am sorry that I forget to revise the title of Figure 3.

Nitrogen content in each part of soybean plants with N0-N0, 15N5-N0, and 15N5-14N5 treatments at 2 and 4 DAT. Average and standard error. Different alphabet on the top of the columns indicates the significantly different in total N among treatments based on Tukey’s Test (P<0.05). Different alphabet at the side of the column indicates the significant difference of N content in the parts among treatments. The alphabet was omitted in the part with no significant difference.

My suggestion:

Nitrogen content in soybean plants treated with 15N label after 2 and 4 days of treatment. The experimental setting is described in the legend of Fig. 1. Different letters indicate significant difference (P<0.5) based on a Tukey’s test. N = (give number of observations) ± SE.

Answer: Thank you very much for good modification. The sentence becomes clearer.

Ln 30-36

Plants absorb N from their roots primarily in the forms of nitrate and ammonium in soils or from fertilizers when applied [1]. Leguminous plants such as soybean can also utilize atmospheric N2 fixed in the root nodule, a symbiotic organ with soil bacteria rhizobia. The nitrate absorbed in the soybean roots is translocated to the shoot in the form of NO3- or it is assimilated in the roots and transported to the leaves mainly in the form of asparagine (Asn) via xylem vessels.

My suggestion:

Plant roots absorb N primarily in the forms of nitrate (NO3-) and ammonium (NH4+). Leguminous plants like soybean, incorporate atmospheric dinitrogen (N2) in root nodules, forming a symbiosis with N2-fixing rhizobia. Nitrate absorbed by soybean roots can directly be translocated to the shoot, or first assimilated and then transported mainly as asparagine (Asn), via xylem vessels to the leaves.

Answer: Thank you very much for your revision.

Ln 97-99

Two 1000 mL polyethylene terephthalate pots were bound, and each pot was independently filled with 800 mL of N-free culture solution covered with aluminum foil with continuous aeration.

My suggestion:

Two polyethylene terephthalate pots with a capacity of 1 L were attached. Each pot was filled with 0.8 L of N-free culture solution, covered with aluminum foil and aerated continuously.

Answer: Thank you very much for your revision.

Answer: For the following items, I revised with your suggestions.

Ln 44 : glutamine (Gln)

Ln 68: glutamate (Glu), aspartate (Asp)

Ln 85-86 My previous comment was not addressed.

under a photon flux density of 228 mmol m-2 s-1 with a 16-hour photoperiod and 8-hour dark period.

The light unit is wrong.

under a photoperiod of 16 h light (228 µmol photons m-2 s-1)/ 8 h darkness

Ln 121: N metabolites instead of N constituents

Reviewer 3 Report

The authors have taken into consideration my comments. The manuscript has been improved.

Author Response

Thank you very much for accepting our revised manuscript. We appreciate your kind comments and suggestions.
